# Oxidative Stress and Negative Consequences on Photosystem II Occasioned by Lead Stress Are Mitigated by 24-Epibrassinolide and Dopamine in Tomato Plants

**DOI:** 10.3390/plants14233699

**Published:** 2025-12-04

**Authors:** Lohana Ribeiro Prestes, Sharon Graziela Alves da Silva, Madson Mateus Santos da Silva, Maria Andressa Fernandes Gonçalves, Elaine Maria Silva Guedes Lobato, Caroline Cristine Augusto, Bruno Lemos Batista, Allan Klynger da Silva Lobato

**Affiliations:** 1Núcleo de Pesquisa Vegetal Básica e Aplicada, Universidade Federal Rural da Amazônia, Rodovia PA 256, Paragominas, Pará 68627-451, Brazil; loohanaribeiroprestes@gmail.com (L.R.P.); sharonalves946@gmail.com (S.G.A.d.S.); engmadsonagro@gmail.com (M.M.S.d.S.); agro.ufra.andressa@gmail.com (M.A.F.G.); elaine.guedes@ufra.edu.br (E.M.S.G.L.); 2Centro de Ciências Naturais e Humanas, Universidade Federal do ABC, Santo André, São Paulo 09280-560, Brazil; augustocc@gmail.com (C.C.A.); bruno.lemos@ufabc.edu.br (B.L.B.)

**Keywords:** brassinosteroids, chlorophyll fluorescence, heavy metal, neurotransmitter, reactive oxygen species, *Solanum lycopersicum*

## Abstract

Food security and human health are directly related to the condition of agricultural soils. Soil contamination by heavy metals is a global environmental problem. Lead (Pb) is a toxic and non-biodegradable element posing a significant risk to ecosystems and human health. 24-Epibrassinolide (EBR) has multiple benefits in plant metabolism, including maximizing gas exchange. In plants, exogenous application of dopamine (DOP) confers tolerance to abiotic stresses, minimizing interferences on growth. This study aimed to investigate whether the exogenous application of EBR and DOP, administered independently or jointly, can contribute to mitigating the oxidative stress and impacts on photosystem II in Pb-stressed tomato, evaluating parameters related to nutritional status, photosystem II activity, gas exchange, antioxidant enzymes, and biomass. Better results were observed with the isolated EBR application, improving the photosynthetic efficiency, as evidenced by the increases in chlorophyll contents, effective quantum yield of PSII photochemistry, photochemical quenching coefficient, and electron transport rate, resulting in a higher net photosynthesis rate. Parallelly, treatment using both plant growth regulators (DOP and EBR) promoted significant increases of 14%, 18%, 13%, and 35% in the activities of superoxide dismutase, catalase, ascorbate peroxidase, and peroxidase, contributing to the reduction in oxidative stress in photosystem II of Pb-stressed plants. Therefore, this research proves that the exogenous application of DOP and EBR, alone or in combination, attenuates the toxic effects generated by Pb in tomato plants.

## 1. Introduction

Tomato (*Solanum lycopersicum* L.) belongs to the Solanaceae family and is a widely cultivated and consumed vegetable crop worldwide [1,2]. It is considered a model plant due to its complete genome sequence, which is widely used in basic and applied research [3,4]. At a global level, it contributes significantly to agricultural trade, representing more than 180 million metric tons produced in more than 161 countries [5]. In terms of production volume among vegetables, it ranks third, surpassed only by potatoes and sweet potatoes [6]. It stands out as the main horticultural product in terms of volume destined for industrial processing [7,8]. It is rich in mineral nutrients such as calcium, phosphorus, magnesium, as well as water-soluble vitamins (B complex and vitamin C), fat-soluble vitamins (A, E, and K), and bioactive compounds that directly contribute to promoting human health [9,10].

Food security and human health are directly related to agriculture conditions [11,12]. Soil contamination by heavy metals is a global environmental issue, posing significant risks, particularly for developing countries [13]. Several studies indicate that anthropogenic activities, such as industrial processes, mining, smelting, oil refineries, paint and pigment manufacturing, metal coatings, steelmaking, thermoelectric plants, and the transportation sector, are potential sources of heavy metal contamination in agricultural ecosystems [14,15,16].

Lead (Pb) is a toxic element that poses a significant risk to human health [17]. Its exposure is associated with damage to the nervous system, especially in children, and is also related to the development of cardiovascular and kidney diseases [18]. In plants, Pb negatively interferes with several physiological processes, such as germination [19], root system development, and nutrient absorption [20]. Additionally, it induces oxidative stress and compromises antioxidant defense mechanisms [21]. These effects culminate in reduced photosynthetic activity [22], impaired general plant metabolism, and, consequently, reduced biomass accumulation and a drop in crop production rates [23].

Plant growth regulators, including brassinosteroids (BRs), play multiple functions in plant metabolism, such as regulating cell division, photosynthesis, reproductive development, enzyme activation, and the synthesis and expression of proteins and genes [24]. These molecules also play a fundamental role in plant tolerance to various types of stress, including biotic and abiotic stress [25], contributing to the mitigation of phytotoxicity caused by heavy metals [26]. Among the BRs, 24-epibrassinolide (EBR) stands out as the most bioactive form naturally synthesized by plants [27,28].

Dopamine (DOP) is a natural catecholamine that acts as a neurotransmitter and neuromodulator in humans and animals, with significant activity also observed in higher plants [29]. In plants, the exogenous application of dopamine has been shown to confer greater tolerance to abiotic stress, promoting increased concentrations of photosynthetic pigments, stomatal conductance, CO_2_ assimilation, and maximum photochemical efficiency [30]. These effects lead to improved plant growth and increased biomass accumulation under adverse conditions [31].

The current literature presents gaps regarding the investigation of the combined action of EBR and DOP in tomato plants under Pb phytotoxicity conditions, evidencing a new technological and pioneering perspective that involves the use of hormonal regulators and plant neurotransmitters. We start from the hypothesis that EBR is capable of positively modulating plant growth and productivity through the regulation of morphological and physiological responses [32,33,34], while DOP contributes to the mitigation of oxidative stress [35,36]. This study aimed to investigate whether the exogenous application of EBR and DOP, administered independently or jointly, can contribute to mitigating the toxic effects of Pb in tomato plants, evaluating parameters related to gas exchange, antioxidant enzyme activity, photosynthetic efficiency and plant development.

## 2. Results

### 2.1. Pb Concentrations Were Minimized After Treatment with DOP and EBR

Treatment with 200 µM Pb caused a significant increase in the contents of this metal in the analyzed tissues (Table 1). However, plants exposed to Pb^2+^ stress and pretreated with DOP had significant reductions of 46% (leaf), 36% (stem) and 30% (root). The use of EBR in stressed plants promoted decreases in leaf (71%), stem (64%) and root (45%), compared to the same treatment in the absence of EBR. The simultaneous use of DOP + EBR promoted substantial attenuations (*p* < 0.05) of 58%, 45% and 42% in leaf, stem and root values, respectively, when compared to the treatment with plants exposed to Pb without pretreatment with DOP or EBR.

### 2.2. DOP and EBR Stimulated the Nutritional Status in Pb-Stressed Plants

Exogenous DOP application increased nutrient levels in these tissues (Table 2), plants treated with DOP exposed to Pb (root) showed increases of 13%, 36%, 17%, 19%, 28% and 8% for Mg, K, Ca, Cu, Zn, and Mn, respectively, comparing to the same treatment without DOP. In the leaves, percentage increases of 15%, 14%, 8%, 27%, 36%, and 22% for Mg, K, Ca, Cu, Zn, and Mn were observed sequentially. Spraying with EBR promoted an increase in nutrient concentrations in these tissues, when compared to plants subjected to the same treatment without EBR. In roots, plants treated with EBR and exposed to Pb presented increases of 34%, 196%, 46%, 24%, 89%, and 29% in the levels of Mg, K, Ca, Cu, Zn, and Mn, respectively (Table 2). In the leaves, the corresponding increases were 41%, 51%, 22%, 48%, 80% and 26% for the same elements. The joint application of DOP and EBR molecules provided increases in nutritional status, being detected increases of 30% (Mg), 104% (K), 39% (Ca), 23% (Cu), 64% (Zn) and 24% (Mn) were observed in the roots (Table 2), when compared to plants subjected to the same treatment without DOP and EBR exposed to Pb. In the leaves, the percentage increases were 18% (Mg), 45% (K), 13% (Ca), 45% (Cu), 61% (Zn), and 25% (Mn) (Table 2).

### 2.3. Plant Growth Regulators Provided Maintenance of Photosynthetic Pigments

Tomato plants exposed to Pb toxicity showed decreases in values of Chl *a*, Chl *b*, Total Chl and Car (Table 3). The application of DOP in plants with Pb excess promoted increases (*p* < 0.05) of 34%, 38%, 35%, and 55% in Chl *a*, Chl *b*, Total Chl and Car, when compared to the Pb^2+^ treatment without DOP. However, the combined Pb^2+^ and EBR treatment promoted increases in Chl *a*, Chl *b*, Total Chl and Car of 42%, 51%, 45% and 76%, respectively, compared to the Pb^2+^ treatment without EBR. The combination of DOP and EBR induced positive effects on the variables Chl *a*, Chl *b*, Chl Total and Car of 36%, 42%, 38% and 67% respectively, when compared to treatments with Pb^2+^ without DOP or EBR.

### 2.4. The Performance of Photosystem II Was Boosted Up After DOP and EBR Spray

Tomato plants subjected to Pb toxicity had an increase in F_0_ value, but reductions in F_v_, F_m_, and F_v_/F_m_ values (Figure 1 and Table 4). However, the exogenous application of growth regulators (EBR and DOP) alleviated the damage on the photosynthetic machinery. In chlorophyll fluorescence (Figure 1 and Table 4), foliar spraying with EBR alone in plants under Pb excess induced a reduction in F_0_ (39%) and significant increases in F_m_ (22%), F_v_ (73%) and F_v_/F_m_ (41%). While the single application of DOP in plants with Pb toxicity reduced F_0_ (35%) and increased the values of F_m_ (12%), F_v_ (50%) and F_v_/F_m_ (35%). On the other hand, the joint application of EBR + DOP decreased F_0_ (39%) and maximized (*p* < 0.05) F_m_ (17%), F_v_ (63%) and F_v_/F_m_ (39%), when compared with the treatment of tomato plants under Pb stress, but without the use of growth regulators. Pb caused decreases in the values of Φ_PSII_, q_P_, ETR, and ETR/*P*_N_, but increased the value of EXC (Table 4). However, the use of EBR promoted increases in Φ_PSII_, q_P_, ETR, and ETR/*P*_N_ of 100%, 42%, 91%, and 31%, respectively, and reduced NPQ and EXC by 28% and 12%, in this order. Similarly, DOP increased the values of Φ_PSII_, q_P_, ETR and ETR/*P*_N_ by 46%. 29%, 43%, and 20%, respectively, and induced reductions in NPQ (7%) and EXC (3%). Used together, EBR and DOP stimulated Φ_PSII_ (77%), q_P_ (31%), ETR (69%), and ETR/*P*_N_ (24%), while decreasing NPQ (23%) and EXC (7%), when compared with the treatment without the use of EBR and DOP and subjected to heavy metal stress.

### 2.5. Exogenous Spraying of DOP and EBR Maximized Gas Exchange

To gas exchange (Table 5), Pb caused deleterious effects. However, the combined treatment of DOP and EBR stimulated increases of 37%, 35%, 66% and 67%, respectively, in *P*_N_, *g*_s_, WUE and *P*_N_/*C*_i_ and a decrease in *C*_i_ (19%), when compared to plants exposed to Pb toxicity without DOP or EBR. The combination of Pb^2+^ with DOP positively altered the variables *P*_N_, *g*_s_, WUE, and *P*_N_/*C*_i_ by 19%, 24%, 35%, and 37%, respectively, compared with plants exposed to Pb^2+^ without DOP. EBR applied in plants with Pb^2+^ increased (*p* < 0.05) 46%, 47%, 80%, and 85% in *P*_N_, *g*_s_, WUE, and *P*_N_/*C*_i_, when compared to the combination without EBR with Pb^2+^.

### 2.6. Neurotransmitter and Brassinosteroids Enhance Antioxidant Defense

Plants exposed to Pb^2+^ intoxication showed increases in the activities of antioxidant enzymes (SOD, CAT, APX and POX) (Figure 2). DOP application promoted significant increases in the enzymatic activities by 7% (SOD), 14% (CAT), 13% (APX) and 31% (POX). Regarding the use of EBR, increases (*p* < 0.05) of 23%, 25%, 31% and 52% were observed for the enzymes SOD, CAT, APX and POX, respectively. However, the treatment with DOP and EBR significantly maximized the enzymatic activities by 14% (SOD), 18% (CAT), 13% (APX) and 35% (POX), compared to the treatment with Pb and without DOP and EBR. Reactive oxygen species (ROS) (H_2_O_2_ and O_2_^−^) and membrane damage indicators (MDA and EL) increased in plants subjected to excess Pb^2+^ (Figure 3). However, plants exposed to Pb^2+^ toxicity and DOP application showed notable attenuations of 23%, 44%, 27% and 24% in H_2_O_2_ and O_2_^−^, MDA and EL, respectively. Plants treated with EBR showed reductions (*p* < 0.05) of 25%, 46%, 29% and 32% in H_2_O_2_, O_2_^−^, MDA and EL, respectively. While the combined use of DOP and EBR resulted in significant reductions of 25%, 45%, 28% and 27% in H_2_O_2_, O_2_^−^, MDA and EL, respectively, compared to the treatment exposed to excess Pb^2+^ and without DOP or EBR.

### 2.7. DOP and EBR Promoted Higher Biomass Accumulation in Plants Exposed to Pb Excess

Stress caused by Pb caused a reduction in biomass (Table 6). Plants exposed to Pb^2+^ stress and pretreated with DOP had significant increases of 50% (LDM), 37% (RDM), 19% (SDM), and 39% (TDM). The use of EBR in stressed plants promoted increases in LDM (50%), RDM (57%), SDM (42%), and TDM (50%), comparing with same treatment in the absence of EBR. The simultaneous use of DOP + EBR promoted substantial increases (*p* < 0.05) of 48%, 48%, 36% and 45% in the values of LDM, RDM, SDM and TDM, respectively, when compared to the treatment with plants exposed to Pb without pretreatment with DOP or EBR.

## 3. Discussion

Overall, the results of this research demonstrate that exposure to Pb caused significant disturbances in plant metabolism, affecting biochemical, nutritional, physiological, and morphological parameters. However, the exogenous application of DOP and EBR, as well as the joint administration of both molecules, attenuated the toxic effects of Pb in tomato plants, synergistically promoting improvements in leaf structure, photosynthetic activity, and plant growth.

DOP, EBR, and combined spray promoted reductions in Pb accumulation levels in the tissues (roots and leaves) of tomato plants. EBR plays a crucial role mitigating Pb toxicity, possibly by regulating ionic homeostasis and maintaining redox balance in plants [37]. DOP acts by increasing the activity of antioxidant enzymes, such as SOD, CAT, APX, and POX, minimizing the accumulation of reactive oxygen species (ROS), such as O_2_, H_2_O_2_, EL, and MDA, favoring *P*_N_ [38,39], corroborated by the results obtained in this study. These effects are in agreement with previous studies that have demonstrated the ability of these phytohormones to modulate antioxidant responses and cellular defense mechanisms against heavy metal stress [40,41,42,43].

Pb transport in tomato plants occurs predominantly through root uptake [44]. In roots, Pb is mobilized into cells through transport proteins, and it is a non-essential ion; there are no specific transporters for Pb [45]. However, many transporters for divalent transition metals (Mg, K, Ca, Cu, Zn, Mn) can facilitate Pb entry through ionic similarity [46]. This interference can be explained due to Pb acting as a competitive analogue, thereby blocking the signal transduction of other nutrients and negatively affecting plant growth and development [47]. EBR treatment significantly reduced Cd accumulation in shoot of rice plants, decreasing the translocation of the metal from root to leaf [48]. Exogenous DOP application (DOP) suppressed the reduction in plant height, root length, chlorophyll levels, antioxidant enzyme activities, and photosynthetic performance in apple seedlings subjected to alkaline stress [38].

Pretreatment with DOP, EBR, and the joint application of both molecules in plants under Pb phytotoxicity conditions promoted positive effects on the levels of macronutrients (Mg, K, Ca) and micronutrients (Cu, Zn, Mn). EBR can mobilize essential minerals, such as manganese (Mn) and zinc (Zn), into the cells. These minerals, acting together, prevent the excessive accumulation of heavy metals, such as lead (Pb), by competing antagonistically for the various transporters and binding sites [49]. In parallel, DOP influences plant growth and development through its interaction with phytohormones, playing a relevant role in the intracellular regulation of ionic permeability and photophosphorylation in chloroplasts. This effect is associated with its high reducing power of this neurotransmitter, which contributes to the elimination of free radicals [50].

The nutrients, including magnesium (Mg), potassium (K) and calcium (Ca) are considered essential secondary macronutrients for the proper functioning of plant metabolism, contributing to increased plant resistance to different types of stress [51]. Calcium (Ca) plays a structural role in the cell wall and membranes, in addition to acting as a secondary messenger in signaling pathways associated with the perception of environmental stimuli, being fundamental in the adaptive responses of plants [52]. Nutritional imbalance, resulting from Mg and K deficiency, has negative repercussions on plant metabolism. K deficiency causes interveinal chlorosis, where the leaves become wrinkled and exhibit upward curling. In severe cases, the shoot tips die. Mg deficiency causes symptoms of leaf yellowing [53].

The micronutrients, such as copper (Cu), zinc (Zn) and manganese (Mn) are accumulated and translocated to different plant organs, such as leaves, roots and stem [54]. Cu acts to establish ideal conditions for plant development, being involved in several physiological processes, including photosynthesis, respiration, hormonal signaling and cellular metabolism [55]. Mn is essential for the functioning of the oxygen evolution complex in photosystem II, participating in water photolysis, a process responsible for releasing of oxygen during photosynthesis [56]. In addition, Mn contributes to the activity of the enzyme superoxide dismutase (SOD), which protects plant cells against oxidative stress by converting reactive oxygen species (ROS) into less toxic [57]. Corroborating our results, application of EBR significantly reduced the concentration of cadmium (Cd) in the roots and shoots of rice plants, attenuating the deleterious effects of Cd stress [58]. In addition, EBR treatment positively influenced the accumulation of iron (Fe) and manganese (Mn), indicating a possible role of these brassinosteroids in the modulation of mineral nutrition under conditions of heavy metal toxicity.

Lead (Pb^2+^) toxicity significantly decreased pigments and carotenoid levels in tomato plants. Pb toxicity impairs photosynthetic attributes by disrupting chloroplasts, causing ultrastructural changes in cells, and negatively affecting thylakoids and pigment synthesis [59]. Conversely, DOP treatment promoted an increase in Chl *a*, Chl *b*, total Chl, and Car levels, which is explained by the neurotransmitter to protect chloroplast structure and reduce damage. The damage caused by the heavy metal is related to the replacement of Mg^2+^, which is a component of the chlorophyll molecule, by Pb^2+^. This substitution was observed in studies with Cd [60,61], these authors showed a decrease in pigments associated with the replacement of Mg^2+^ by the heavy metal in the tetrapyrrole centers of chlorophyll, which causes the breakdown of the molecule and irreversible modifications of LHCII. The application of EBR improved the properties of leaf pigments by reducing chlorophyll degradation, corroborating the reduction in MDA and EL levels observed in this research. Carotenoids are responsible for protecting chlorophyll from oxidative damage by scavenging ROS, minimizing negative consequences on structure of the photosystems [62]. Research evaluating tomato plants, concluded that the application of EBR caused increases in carotenoid values, alleviating Cr toxicity [63]. A study demonstrated that apple plants under nutritional stress showed smaller declines in Chl *a*, Chl *b*, and total Chl levels in plants under DOP application [64].

Exogenous application of DOP and EBR increased Φ_PSII_, q_P_, ETR, and ETR/*P*_N_ values, demonstrating that these growth regulators, used alone and in combination, reduced the damage caused by Pb toxicity by boosting electron transport and increasing the efficiency of photosynthetic activity. Plants under Pb stress exhibit decreased photosynthetic activity as a consequence of reduced CO_2_ assimilation, electron transport, inhibition of reaction centers, and photochemical limitation [65]. In contrast, EBR and DOP reduced NPQ and EXC values, confirming that these molecules acted to reduce the accumulation of excess energy and non-photochemical dissipation, favoring the energy availability for the photosynthetic machinery [66]. Improvements in photosynthesis and fluorescence parameters when analyzing cucumber plants pretreated with 150 μM DOP and stressed with 500 μmol L^−1^ nitrate, with increases in F_v_/F_m_, q_P_, and ETR [67]. Similarly, the study investigating the effects of EBR on Cr-intoxicated tomato plants, demonstrated positive results in chlorophyll content, specifically with increases of 53%, 29%, and 44% in F_v_/F_m_, Φ_PSII_, and q_P_, respectively [63].

Plants treated with DOP and EBR and subjected to 200 µM Pb^2+^ toxicity showed increases in CO_2_ fixation. Pb toxicity causes a reduction in gas exchange and induces stomatal closure, which limits CO_2_ entry into cells and water supply, as confirmed by the decrease in *g*_s_ and *E*, negatively affecting *P*_N_ [68,69]. Ref. [70] evaluating Pb-intoxicated cucumber plants, observed increases in *C*_i_ and decreases in *P*_N_ and *g*_s_, corroborating this study. DOP clearly attenuated the negative effects of Pb, promoting improved water content and absorption, which can be explained by the increase in WUE [50]. Increases in *P*_N_ and *P*_N_//*C*_i_ were identified in plants treated with EBR, proving that this molecule facilitates the absorption and flow of CO_2_ in the intercellular spaces, maximizing the activity of the RuBisCO enzyme, intrinsically linked to CO_2_ assimilation [71,72]. Effects associated with utilization of 100 μM DOP in apple plants under Cd stress promoted increases in *E*, *g*_s_, and *P*_N_, while *C*_i_ had a reduction, delaying the impact caused by this heavy metal on growth and development [35]. *E. urophylla* plants treated with EBR attenuated the toxic effects of Cd^2+^ on gas exchange, contributing to increases of 21% in WUE and 23% in *P*_N_//*C*_i_ [73].

Plants exposed to Pb^2+^ toxicity and treated with DOP and EBR showed increases in the activities of antioxidant enzymes (SOD, CAT, APX, and POX). These results can be justified by the action of DOP and EBR that influence the defense mechanisms in plants, boosting the enzymatic activities linked to redox metabolism, attenuating the imbalance caused by the high concentration of ROS in the tissues and damage to cell membranes [74]. The enzymes that perform antioxidant functions in metabolism are SOD (a key enzyme), which accelerates the reduction reaction of O_2_^−^ to H_2_O_2_ and O_2_. CAT, APX, and POX reduce H_2_O_2_ to H_2_O or other molecules that are beneficial to plant cells and, consequently to metabolism [75]. The results obtained corroborate previous research, who investigated the contributions of DOP in cucumber plants under nitrate stress, stimulating the activities of antioxidant enzymes [76]. In general, ROS impair cellular components and disrupt the cytoplasmic balance responsible for signaling the antioxidant defense system and eliminating O_2_^−^ and H_2_O_2_ [77]. However, our studies reveal that DOP and EBR mitigated the negative impacts caused by Pb^2+^ intoxication, possibly by regulating carbon utilization and hormonal signaling. Mechanisms of action of EBR aiming to alleviate the harmful effects caused by Cr, being detected increases in activities of SOD (27%), CAT (19%) and APX (107%), reducing oxidative stress, confirming the results of this research [63].

Reductions in ROS concentrations and membrane damage were found in plants treated with DOP and EBR, indicating that these molecules attenuated the toxic effects of ROS. This study demonstrated that the leaf tissues of plants subjected to Pb^2+^ poisoning suffered less oxidative damage (MDA and EL), these results being explained by the increases in antioxidant enzyme activities (POX, SOD, CAT and APX), previously described. Heavy metal toxicity can cause deleterious effects on plant cells, associated with the excess accumulation of O_2_^−^ and H_2_O_2_, causing damage to cellular structures, such as membrane rupture, protein denaturation and reduced photochemical activity [78]. A study with *Festuca arundinacea* plants treated with EBR and under Pb stress, describing a 10% reduction in O_2_^−^ concentrations [79]. Corroborating the results found in this research, tomato plants grown under chromium intoxication had an increase of 118% in EL. However, an exogenous application of DOP (0.1 mM) significantly reduced EL, preventing damage to cell membranes [29].

DOP and EBR positively regulate the biomass of Pb-stressed tomato plants through increases in LDM, SDM, RDM, and TDM, mitigating the toxic effects of this heavy metal. Exogenous DOP application mitigates Pb stress by increasing pigment concentrations, photochemical activity, and carbon fixation. These effects contribute to stimulating growth and favoring biomass accumulation [30]. Ref. [64] detected that DOP is capable of increasing stem height, promoting higher biomass accumulation in apple plants under nutritional stress conditions. EBR reduced the Pb uptake by altering cell membrane permeability and stimulating the production of enzymes linked to plant defense system [80]. Ref. [81] also observed that Zn-stressed soybean plants treated with EBR showed increased biomass. Ref. [73] observed that EBR application improved the biomass of *Eucalyptus urophylla* plants exposed to Cd toxicity. Pb excess within cells causes chloroplast disorganization and leaf damage, including ultrastructural lesions and even cell death. Furthermore, excess Pb impairs chlorophyll production and can interfere with nutrient transport and photosynthesis [82]. Studies such as that by [83] show that increased Pb application decreases biomass of bean plants.

## 4. Materials and Methods

### 4.1. Location and Growth Conditions

The experiment was performed at the Campus of Paragominas of the Universidade Federal Rural da Amazônia, Paragominas, Brazil (2°55′ S, 47°34′ W). The study was conducted in a greenhouse with controlled temperature and humidity. The minimum, maximum, and median temperatures were 23.6, 31.7 and 26.8 °C, respectively. The relative humidity during the experimental period varied between 60% and 80%, with photosynthetic photon flux density (PPFD) oscillating between 1150 and 550 µmol m^−2^ s^−1^ (measured at 12 am), and 12-h photoperiod.

### 4.2. Plants, Containers, and Acclimation

Seeds of *Solanum lycopersicum* L. cv Santa Clara were germinated using vegetable substrate and transplanted on the 15th day into 1.2-L pots filled with a mixed substrate of sand and vermiculite at a ratio of 3:1. Plants were cultivated under semi-hydroponic conditions containing 500 mL of nutritive solution. A nutritive solution was used as a source of nutrients [84]; the ionic strength started at 50% and was modified to 100% after two days.

### 4.3. Experimental Design

Experimental design had eight randomized treatments: Control − DOP − EBR (1), control − DOP + EBR (2), control + DOP − EBR (3), control + DOP + EBR (4), Pb^+2^ − DOP − EBR (5), Pb^+2^ − DOP + EBR (6), Pb^+2^ + DOP − EBR (7) and Pb^+2^ + DOP + EBR (8). Five replicates for each one of the eight treatments were conducted and used in the experiment, a total of 40 experimental units, with one plant in each unit.

### 4.4. Dopamine (DOP) and 24-Epibrassinolide (EBR) Preparations and Applications

Dopamine (DOP) solution with 100 µM (Sigma-Aldrich™, Saint Louis, MO, USA) was prepared [85]. A 100 nM 24-epibrassinolide (EBR) (Sigma-Aldrich™, Saint Louis, MO, USA) solution was used [86]. Fifteen-day-old plants were sprayed with single and combined applications of DOP and EBR (pre-treatment) at 7-d intervals until day 50.

### 4.5. Plant Nutrition and Pb Treatment

The plants received the following macro and micronutrients contained in the nutrient solution [87]: 8.75 mM KNO_3_, 7.5 mM Ca(NO_3_)_2_·4H_2_O, 3.25 mM NH_4_H_2_PO_4_, 1.5 mM MgSO_4_·7H_2_O, 62.50 µM KCl, 31.25 µM H_3_BO_3_, 2.50 µM MnSO_4_·H_2_O, 2.50 µM ZnSO_4_·7H_2_O, 0.63 µM CuSO_4_·5H_2_O, 0.63 µM NaMoO_4_·5H_2_O, and 250.0 µM NaEDTAFe·3H_2_O. To induce Pb stress, PbCl_2_ was used at concentrations of 0 and 200 µM Pb and was applied over 10 days (days 40–50 after the start of the experiment). During the study, the nutrient solutions were changed at 07:00 h at 3-day intervals, with the pH adjusted to 5.5 using HCl or NaOH. On day 50 of the experiment, physiological and morphological parameters were measured for all plants, and tissues were harvested for anatomical, biochemical and nutritional analyses.

### 4.6. Pb Determination

Milled samples (100 mg) of leaf tissue were pre-digested in conical tubes (50 mL) with 2 mL of sub-boiled HNO_3_. Subsequently, 8 mL of a solution containing 4 mL of H_2_O_2_ (30% *v*/*v*) and 4 mL of ultra-pure water were added and transferred to a Teflon digestion vessel [88]. Determination of Pb was performed using an inductively coupled plasma mass spectrometer (model ICP-MS 7900; Agilent, Santa Clara, CA, USA).

### 4.7. Measurement of Chlorophyll Fluorescence

The minimal fluorescence yield of the dark-adapted state (F_0_), maximal fluorescence yield of the dark-adapted state (F_m_), variable fluorescence (F_v_), maximal quantum yield of PSII photochemistry (F_v_/F_m_), effective quantum yield of PSII photochemistry (Φ_PSII_), photochemical quenching coefficient (q_P_), nonphotochemical quenching (NPQ), electron transport rate (ETR), relative energy excess at the PSII level (EXC), and ratio between the electron transport rate and net photosynthetic rate (ETR/*P*_N_) were determined using a modulated chlorophyll fluorometer (model OS5p; Opti-Sciences, Hudson, NH, USA). The chlorophyll fluorescence was measured in fully expanded leaves under light between 10:00 and 12:00 h. Preliminary tests determined that the acropetal third of leaves in the middle third of the plant and that adapted to the dark for 30 min yielded the greatest F_v_/F_m_ ratio. Therefore, this part of the plant was used for measurements. The intensity and duration of the saturation light pulse were 7500 µmol m^–2^ s^–1^ and 0.7 s, respectively.

### 4.8. Evaluation of Gas Exchange

The net photosynthetic rate (*P*_N_), transpiration rate (*E*), stomatal conductance (*g*_s_), and intercellular CO_2_ concentration (*C*_i_) were evaluated using an infrared gas analyser (model LCPro^+^; ADC BioScientific, Hoddesdon, UK). These parameters were measured at the adaxial surface of fully expanded leaves that were collected from the middle region of the plant. The water-use efficiency (WUE) was determined [89] and the instantaneous carboxylation efficiency (*P*_N_/*C*_i_) was calculated [90]. Gas exchange was evaluated in all plants under a constant CO_2_ concentration (360 μmol mol^−1^ CO_2_), photosynthetically active radiation (800 μmol photons m^−2^ s^−1^), air-flow rate (300 µmol s^−1^), and temperature (28 °C) in the test chamber between 10:00 and 12:00 h.

### 4.9. Extraction of Antioxidant Enzymes, Superoxide and Soluble Proteins

Antioxidant enzymes (SOD, CAT, APX, and POX), superoxide, and soluble proteins were extracted from leaf tissues [91]. The extraction mixture was prepared by homogenizing 500 mg of fresh plant material in 5 mL of extraction buffer, which consisted of 50 mM phosphate buffer (pH 7.6), 1.0 mM ascorbate, and 1.0 mM EDTA. Samples were centrifuged at 14,000× *g* for 4 min at 3 °C, and the supernatant was collected. Quantification of the total soluble proteins was performed [92]. Absorbance was measured at 595 nm, using bovine albumin as a standard.

### 4.10. Superoxide Dismutase Assay

For the SOD assay (EC 1.15.1.1), 2.8 mL of a reaction mixture containing 50 mM phosphate buffer (pH 7.6), 0.1 mM EDTA, 13 mM methionine (pH 7.6), 75 µM NBT, and 4 µM riboflavin was mixed with 0.2 mL of supernatant. The absorbance was then measured at 560 nm [93]. One SOD unit was defined as the amount of enzyme required to inhibit 50% of the NBT photoreduction. The SOD activity was expressed in unit mg^–1^ protein.

### 4.11. Catalase Assay

For the CAT assay (EC 1.11.1.6), 0.2 mL of supernatant and 1.8 mL of a reaction mixture containing 50 mM phosphate buffer (pH 7.0) and 12.5 mM hydrogen peroxide were mixed, and the absorbance was measured at 240 nm [94].The CAT activity was expressed in μmol H_2_O_2_ mg^–1^ protein min^–1^.

### 4.12. Ascorbate Peroxidase Assay

For the APX assay (EC 1.11.1.11), 1.8 mL of a reaction mixture containing 50 mM phosphate buffer (pH 7.0), 0.5 mM ascorbate, 0.1 mM EDTA, and 1.0 mM hydrogen peroxide was mixed with 0.2 mL of supernatant and the absorbance was measured at 290 nm [95]. The APX activity was expressed in μmol AsA mg^–1^ protein min^–1^.

### 4.13. Peroxidase Assay

For the POX assay (EC 1.11.1.7), 1.78 mL of a reaction mixture containing 50 mM phosphate buffer (pH 7.0) and 0.05% guaiacol was mixed with 0.2 mL of supernatant, followed by the addition of 20 µL of 10 mM hydrogen peroxide. The absorbance was then measured at 470 nm [96]. The POX activity was expressed in μmol tetraguaiacol mg^–1^ protein min^–1^.

### 4.14. Determination of Superoxide Concentration

For the determination of O_2_^−^, 1 mL of extract was incubated with 30 mM phosphate buffer [pH 7.6] and 0.51 mM hydroxylamine hydrochloride for 20 min at 25 °C. Seventeen mM sulphanilamide and 7 mM α-naphthylamine were then added to the incubation mixture for 20 min at 25 °C. After the reaction, ethyl ether was added in the identical volume and centrifuged at 3000× *g* for 5 min. The absorbance was measured at 530 nm [97].

### 4.15. Extraction of Nonenzymatic Compounds

Nonenzymatic compounds (H_2_O_2_ and MDA) were extracted [98]. Briefly, a mixture designed to extract H_2_O_2_ and MDA was prepared by homogenizing 500 mg of fresh leaf material in 5 mL of 5% (*w*/*v*) trichloroacetic acid. Samples were then centrifuged at 15,000× *g* for 15 min at 3 °C to collect the supernatant.

### 4.16. Determination of Hydrogen Peroxide Concentration

To measure H_2_O_2_, 200 µL of supernatant and 1800 µL of reaction mixture (2.5 mM potassium phosphate buffer [pH 7.0] and 500 mM potassium iodide) were mixed and the absorbance was measured at 390 nm [99].

### 4.17. Quantification of Malondialdehyde Concentration

MDA was determined by mixing 500 µL of supernatant with 1 mL of the reaction mixture, which contained 0.5% (*w*/*v*) thiobarbituric acid in 20% trichloroacetic acid. The mixture was incubated in boiling water at 95 °C for 20 min with the reaction terminated by placing the reaction container in an ice bath. The samples were centrifuged at 10,000× g for 10 min and the absorbance was measured at 532 nm. The nonspecific absorption at 600 nm was subtracted from the absorbance data. The amount of MDA–TBA complex (red pigment) was calculated [100], with minor modifications and using an extinction coefficient of 155 mM^−1^ cm^−1^.

### 4.18. Determination of Electrolyte Leakage

Electrolyte leakage was measured with minor modifications [101]. Fresh tissue (200 mg) was cut into pieces 1 cm in length and placed in containers with 8 mL of distilled deionized water. The containers were incubated in a water bath at 40 °C for 30 min and the initial electrical conductivity of the medium (EC_1_) was measured. The samples were then boiled at 95 °C for 20 min to release the electrolytes. After cooling, the final electrical conductivity (EC_2_). The percentage of electrolyte leakage was calculated using the formula EL (%) = (EC_1_/EC_2_) × 100.

### 4.19. Determination of Photosynthetic Pigments

Determinations of the chlorophyll and carotenoid levels were performed with 40 mg of leaf tissue. The samples were homogenized in the dark with 8 mL of 90% methanol (Sigma-Aldrich™, Saint Louis, MO, USA). The homogenate was centrifuged at 6000× *g* for 10 min at 5 °C. The supernatant was removed and the chlorophyll *a* (Chl *a*) and *b* (Chl *b*), carotenoid (Car), and total chlorophyll (total Chl) levels were quantified using a spectrophotometer (model UV-M51; Bel Photonics, Monza, Italy) [102].

### 4.20. Measurements of Biomass

The biomass of roots and leaves was measured based on constant dry weights (g) after drying in a forced-air ventilation oven at 65 °C.

### 4.21. Data Analysis

The data were subjected to an analysis of variance, and significant differences between the means were determined using the Scott-Knott test at a probability level of 5%, because this test better distinguishes the differences between averages [103]. Standard deviations were calculated for each treatment. Statistical analysis of the data was done using R™ software version 4.4.1 [104].

## 5. Conclusions

This study confirmed that the application of EBR (24-epibrassinolide) and DOP (dopamine), alone or in combination, mitigated the damage caused by Pb (lead) exposure in tomato plants. However, the most pronounced effects were observed with the isolated application of EBR. The exogenous application of EBR attenuated photoinhibition and contributed to the maintenance of photosynthetic efficiency, as evidenced by the increases in chlorophyll contents, effective quantum yield of PSII photochemistry, photochemical quenching coefficient, and electron transport rate, resulting in a higher net photosynthesis rate. Additionally, the defense system was enhanced by EBR and DOP treatments, indicating a key role in mitigating oxidative damage induced by Pb excess. These treatments clearly protected the photosynthetic machinery, upregulating the antioxidant system and improving gas exchange. Both treatments alleviated Pb-induced oxidative stress by stimulating enzymes linked to redox metabolism, detected with superoxide dismutase, catalase, ascorbate peroxidase, and peroxidase, which are involved in the detoxification of reactive oxygen species, including superoxide anion and hydrogen peroxide, in Pb-stressed plants. Simultaneously, the combination of DOP and EBR promoted synergistic effects, favoring the antioxidant system, the integrity of the photosynthetic machinery, and gas exchange, with positive repercussions on improved nutritional status and biomass accumulation, as verified in the results obtained. Therefore, this study demonstrated that EBR and DOP alleviated the negative interferences caused by Pb stress in tomato plants. However, it is recommended that future studies include molecular approaches to determine the modulated gene expression of DOP and EBR in plants under Pb toxicity.

## Figures and Tables

**Figure 1 plants-14-03699-f001:**
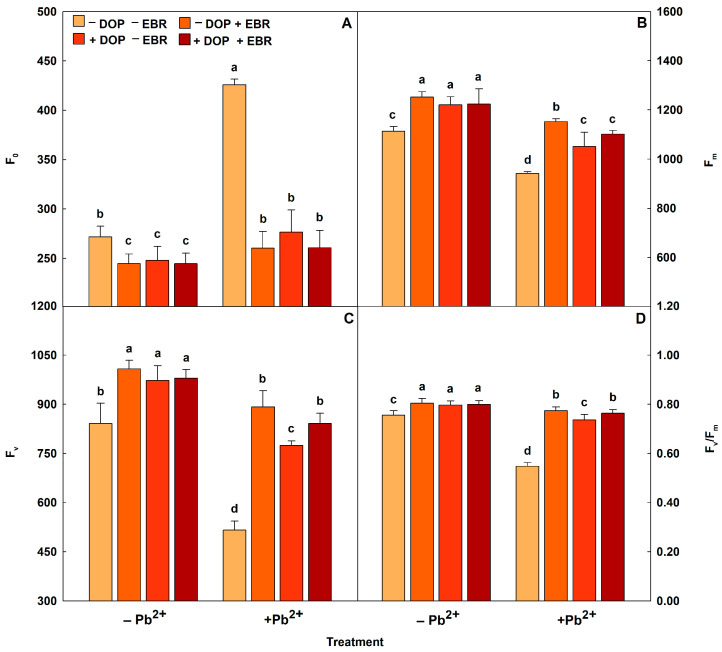
Minimal fluorescence yield of the dark-adapted state (F_0_), maximal fluorescence yield of the dark-adapted state (F_m_), variable fluorescence (F_v_) and maximal quantum yield of PSII photochemistry (F_v_/F_m_) in tomato plants treated with 24-epibrassinolide (EBR) and subjected to lead (Pb) toxicity. Columns with different letters indicate significant differences from the Scott-Knott test (*p* < 0.05). Means ± SD, n = 5. (**A**) F_0_, (**B**) F_m_, (**C**) F_v_, (**D**) F_v_/F_m_.

**Figure 2 plants-14-03699-f002:**
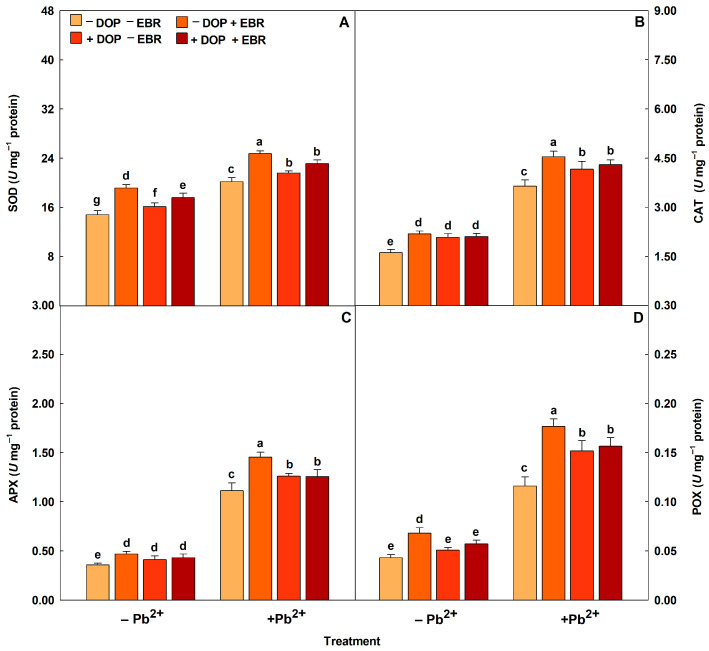
Activities of superoxide dismutase (SOD), catalase (CAT), ascorbate peroxidase (APX) and peroxidase (POX) in tomato plants treated with 24-epibrassinolide (EBR) and subjected to lead (Pb) toxicity. Columns with different letters indicate significant differences from the Scott-Knott test (*p* < 0.05). Means ± SD, n = 5. (**A**) SOD, (**B**) CAT, (**C**) APX, (**D**) POX.

**Figure 3 plants-14-03699-f003:**
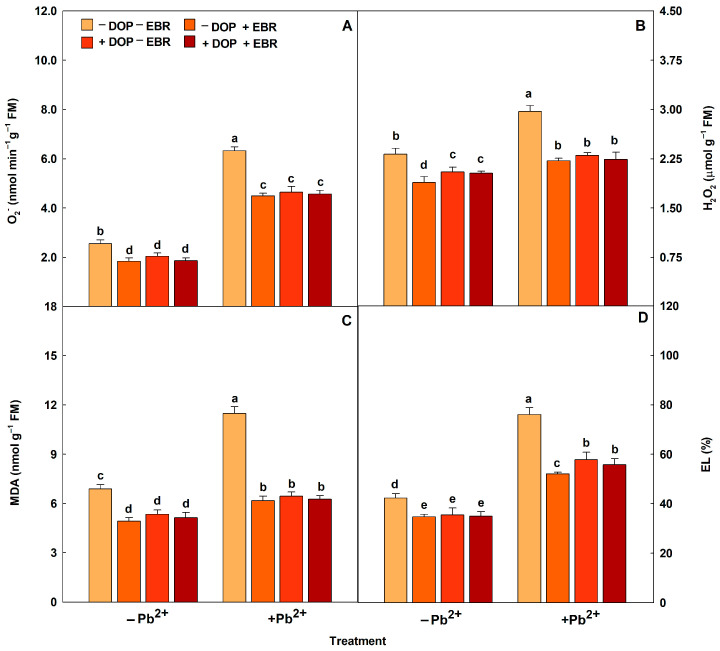
Superoxide (O_2_^−^), hydrogen peroxide (H_2_O_2_), malondialdehyde (MDA) and electrolyte leakage (EL) in tomato plants treated with 24-epibrassinolide (EBR) and subjected to lead (Pb) toxicity. Columns with different letters indicate significant differences from the Scott-Knott test (*p* < 0.05). Means ± SD, n = 5. (**A**) O_2_^−^, (**B**) H_2_O_2_, (**C**) MDA, (**D**) EL.

**Table 1 plants-14-03699-t001:** Pb contents in tomato plants treated with EBR and subjected to Pb toxicity.

Pb^2+^	DOP	EBR	Pb in Leaf (µg g DM^−1^)	Pb in Stem (µg g DM^−1^)	Pb in Root (µg g DM^−1^)
−	−	−	0.000 ± 0.000 d	0.000 ± 0.000 d	0.000 ± 0.000 d
−	−	+	0.000 ± 0.000 d	0.000 ± 0.000 d	0.000 ± 0.000 d
−	+	−	0.000 ± 0.000 d	0.000 ± 0.000 d	0.000 ± 0.000 d
−	+	+	0.000 ± 0.000 d	0.000 ± 0.000 d	0.000 ± 0.000 d
+	−	−	0.106 ± 0.004 a	0.011 ± 0.001 a	0.132 ± 0.008 a
+	−	+	0.031 ± 0.007 d	0.004 ± 0.001 c	0.073 ± 0.003 c
+	+	−	0.057 ± 0.004 b	0.007 ± 0.001 b	0.093 ± 0.003 b
+	+	+	0.044 ± 0.003 c	0.006 ± 0.001 b	0.076 ± 0.006 c

Pb = lead; DOP = dopamine; EBR = 24-epibrassinolide. Columns with different letters indicate significant differences from the Scott-Knott test (*p* < 0.05). Means ± SD, n = 5.

**Table 2 plants-14-03699-t002:** Nutrient contents in tomato plants treated with EBR and subjected to Pb toxicity.

Pb^2+^	DOP	EBR	Mg (mg g DM^−1^)	K (mg g DM^−1^)	Ca (mg g DM^−1^)	Cu (µg g DM^−1^)	Zn (µg g DM^−1^)	Mn (µg g DM^−1^)
Contents in root								
−	−	−	6.84 ± 0.26 c	35.22 ± 3.04 d	3.50 ± 0.15 c	10.17 ± 0.10 d	33.81 ± 0.94 c	118.82 ± 2.18 d
−	−	+	8.38 ± 0.22 a	47.70 ± 2.35 a	5.52 ± 0.17 a	13.45 ± 0.20 a	41.19 ± 1.42 a	174.52 ± 5.13 a
−	+	−	7.42 ± 0.12 b	38.27 ± 0.21 c	4.51 ± 0.18 b	11.49 ± 0.27 c	36.34 ± 1.66 b	143.84 ± 4.41 c
−	+	+	7.54 ± 0.26 b	41.05 ± 1.73 b	5.34 ± 0.20 a	12.55 ± 0.30 b	37.14 ± 1.57 b	163.77 ± 1.76 b
+	−	−	3.31 ± 0.15 f	0.83 ± 0.08 e	1.57 ± 0.10 f	7.06 ± 0.53 f	9.60 ± 1.54 g	44.94 ± 1.95 f
+	−	+	4.45 ± 0.15 d	2.46 ± 0.13 e	2.29 ± 0.17 d	8.73 ± 0.17 e	18.17 ± 0.95 d	57.95 ± 2.13 e
+	+	−	3.75 ± 0.28 e	1.13 ± 0.07 e	1.83 ± 0.07 e	8.40 ± 0.12 e	12.31 ± 0.69 f	48.53 ± 2.26 f
+	+	+	4.31 ± 0.14 d	1.69 ± 0.12 e	2.19 ± 0.16 d	8.71 ± 0.15 e	15.75 ± 1.04 e	55.88 ± 1.72 e
Contents in leaf								
−	−	−	6.40 ± 0.25 d	46.16 ± 2.24 b	13.24 ± 0.90 c	8.27 ± 0.09 d	30.78 ± 1.60 c	46.33 ± 1.33 c
−	−	+	8.36 ± 0.08 a	55.01 ± 2.62 a	18.62 ± 0.97 a	10.50 ± 0.21 a	37.58 ± 1.37 a	58.25 ± 1.72 a
−	+	−	7.17 ± 0.14 c	52.58 ± 0.66 a	15.80 ± 0.84 b	9.27 ± 0.15 c	34.55 ± 2.05 b	51.33 ± 1.06 b
−	+	+	7.52 ± 0.26 b	53.94 ± 1.65 a	17.82 ± 0.83 a	9.63 ± 0.11 b	35.78 ± 1.36 b	56.52 ± 0.69 a
+	−	−	5.53 ± 0.28 e	31.03 ± 1.79 d	11.00 ± 1.00 f	5.53 ± 0.04 f	7.71 ± 1.54 f	37.95 ± 1.91 d
+	−	+	7.78 ± 0.37 b	46.91 ± 1.62 b	13.45 ± 0.72 d	8.18 ± 0.45 d	13.91 ± 1.33 d	47.96 ± 2.33 c
+	+	−	6.34 ± 0.14 d	35.43 ± 1.93 c	11.91 ± 0.86 e	7.01 ± 0.13 e	10.49 ± 1.13 e	46.36 ± 1.15 c
+	+	+	6.52 ± 0.39 d	44.84 ± 1.39 b	12.48 ± 1.05 d	8.02 ± 0.38 d	12.40 ± 0.96 d	47.26 ± 1.40 c

Pb = lead; DOP = dopamine; EBR = 24-epibrassinolide; Mg = magnesium; K = potassium; Ca = calcium; Cu = copper; Zn = zinc; Mn = manganese. Columns with different letters indicate significant differences from the Scott-Knott test (*p* < 0.05). Means ± SD, n = 5.

**Table 3 plants-14-03699-t003:** Photosynthetic pigments in tomato plants treated with EBR and subjected to Pb toxicity.

Pb^2+^	DOP	EBR	Chl *a* (mg g^–1^ FM)	Chl *b* (mg g^–1^ FM)	Total Chl (mg g^–1^ FM)	Car (mg g^–1^ FM)	Ratio Chl *a*/Chl *b*	Ratio Total Chl/Car
−	−	−	9.04 ± 0.26 c	5.48 ± 0.11 c	14.52 ± 0.17 d	0.74 ± 0.03 c	1.65 ± 0.08 b	19.54 ± 0.81 d
−	−	+	10.20 ± 0.16 a	7.24 ± 0.21 a	17.44 ± 0.15 a	0.96 ± 0.02 a	1.41 ± 0.06 c	18.14 ± 0.47 e
−	+	−	9.72 ± 0.13 b	6.70 ± 0.16 b	16.42 ± 0.20 c	0.89 ± 0.02 b	1.45 ± 0.04 c	18.42 ± 0.25 e
−	+	+	9.92 ± 0.19 b	6.90 ± 0.07 b	16.82 ± 0.22 b	0.92 ± 0.02 a	1.44 ± 0.03 c	18.21 ± 0.31 e
+	−	−	5.70 ± 0.23 f	3.08 ± 0.13 f	8.78 ± 0.26 g	0.33 ± 0.1 f	1.85 ± 0.12 a	26.46 ± 0.82 a
+	−	+	8.08 ± 0.20 d	4.66 ± 0.11 d	12.74 ± 0.15 e	0.58 ± 0.02 d	1.74 ± 0.08 b	22.07 ± 0.88 c
+	+	−	7.63 ± 0.12 e	4.24 ± 0.11 e	11.87 ± 0.20 f	0.51 ± 0.01 e	1.80 ± 0.04 a	23.28 ± 0.39 b
+	+	+	7.76 ± 0.10 e	4.36 ± 0.13 e	12.12 ± 0.22 f	0.55 ± 0.04 e	1.78 ± 0.04 a	22.20 ± 1.45 c

Pb = lead; DOP = dopamine; EBR = 24-epibrassinolide; Chl *a* = chlorophyll *a*; Chl *b* = chlorophyll *b*; Total chl = total chlorophyll; Car = carotenoids. Columns with different letters indicate significant differences from the Scott-Knott test (*p* < 0.05). Means ± SD, n = 5.

**Table 4 plants-14-03699-t004:** Chlorophyll fluorescence in tomato plants treated with DOP and EBR and subjected to Pb toxicity.

Pb^2+^	DOP	EBR	Φ_PSII_	q_P_	NPQ	ETR (µmol m^−2^ s^−1^)	EXC (µmol m^−2^ s^−1^)	ETR/*P*_N_
−	−	−	0.27 ± 0.01 c	0.71 ± 0.03 c	0.32 ± 0.08 c	40.54 ± 2.44 d	0.63 ± 0.02 c	2.83 ± 0.08 b
−	−	+	0.41 ± 0.02 a	0.89 ± 0.02 a	0.15 ± 0.01 e	60.58 ± 3.49 a	0.49 ± 0.03 e	3.52 ± 0.23 a
−	+	−	0.35 ± 0.01 b	0.79 ± 0.02 b	0.22 ± 0.04 d	52.56 ± 2.48 c	0.55 ± 0.02 d	3.35 ± 0.21 a
−	+	+	0.38 ± 0.02 a	0.87 ± 0.05 a	0.18 ± 0.01 d	56.56 ± 3.38 b	0.51 ± 0.03 e	3.40 ± 0.22 a
+	−	−	0.13 ± 0.01 f	0.48 ± 0.01 e	0.57 ± 0.09 a	20.00 ± 1.71 g	0.75 ± 0.01 a	2.17 ± 0.18 c
+	−	+	0.26 ± 0.01 c	0.68 ± 0.03 c	0.41 ± 0.06 b	38.22 ± 1.79 d	0.66 ± 0.01 c	2.85 ± 0.10 b
+	+	−	0.19 ± 0.01 e	0.62 ± 0.04 d	0.53 ± 0.08 a	28.52 ± 1.67 f	0.73 ± 0.02 a	2.61 ± 0.22 b
+	+	+	0.23 ± 0.01 d	0.63 ± 0.02 d	0.44 ± 0.05 b	33.82 ± 2.08 e	0.70 ± 0.02 b	2.69 ± 0.21 b

Pb = lead; DOP = dopamine; EBR = 24-epibrassinolide; Φ_PSII_ = effective quantum yield of PSII photochemistry; q_P_ = photochemical quenching coefficient; NPQ = nonphotochemical quenching; ETR = electron transport rate; EXC = relative energy excess at the PSII level; ETR/*P*_N_ = ratio between the electron transport rate and net photosynthetic rate. Columns with different letters indicate significant differences from the Scott-Knott test (*p* < 0.05). Means ± SD, n = 5.

**Table 5 plants-14-03699-t005:** Gas exchange in tomato plants treated with EBR and subjected to Pb toxicity.

Pb^2+^	DOP	EBR	*P*_N_ (µmol m^−2^ s^−1^)	*E* (mmol m^−2^ s^−1^)	*g*_s_ (mol m^−2^ s^−1^)	*C*_i_ (µmol mol^−1^)	WUE (µmol mmol^–1^)	*P*_N_/*C*_i_ (µmol m^−2^ s^−1^ Pa^−1^)
−	−	−	14.30 ± 0.53 c	2.87 ± 0.10 d	0.38 ± 0.01 b	308 ± 9 b	4.99 ± 0.28 d	0.046 ± 0.003 d
−	−	+	17.22 ± 0.50 a	2.72 ± 0.09 d	0.43 ± 0.01 a	224 ± 10 d	6.35 ± 0.22 a	0.077 ± 0.005 a
−	+	−	15.66 ± 0.51 b	2.83 ± 0.14 d	0.40 ± 0.02 b	278 ± 11 c	5.53 ± 0.19 c	0.056 ± 0.002 c
−	+	+	16.62 ± 0.31 a	2.79 ± 0.14 d	0.41 ± 0.02 a	239 ± 6 d	5.96 ± 0.22 b	0.070 ± 0.003 b
+	−	−	9.17 ± 0.16 g	3.77 ± 0.08 a	0.17 ± 0.01 e	344 ± 19 a	2.43 ± 0.03 h	0.027 ± 0.001 f
+	−	+	13.40 ± 0.38 d	3.06 ± 0.09 c	0.25 ± 0.01 c	271 ± 9 c	4.38 ± 0.22 e	0.050 ± 0.003 d
+	+	−	10.92 ± 0.46 f	3.33 ± 0.04 b	0.21 ± 0.01 d	297 ± 12 b	3.28 ± 0.14 g	0.037 ± 0.003 e
+	+	+	12.56 ± 0.41 e	3.11 ± 0.06 c	0.23 ± 0.01 c	277 ± 9 c	4.04 ± 0.16 f	0.045 ± 0.002 d

Pb = lead; DOP = dopamine; EBR = 24-epibrassinolide; *P*_N_ = net photosynthetic rate; *E* = transpiration rate; *g*_s_ = stomatal conductance; *C*_i_ = intercellular CO_2_ concentration; WUE = water-use efficiency; *P*_N_/*C*_i_ = carboxylation instantaneous efficiency. Columns with different letters indicate significant differences from the Scott-Knott test (*p* < 0.05). Means ± SD, n = 5.

**Table 6 plants-14-03699-t006:** Biomass of tomato plants treated with EBR and subjected to Pb toxicity.

Pb^2+^	DOP	EBR	LDM (g plant^−1^)	RDM (g plant^−1^)	SDM (g plant^−1^)	TDM (g plant^−1^)
−	−	−	6.47 ± 0.28 d	2.20 ± 0.04 e	2.50 ± 0.07 d	11.17 ± 0.36 f
−	−	+	7.62 ± 0.07 a	3.20 ± 0.14 a	3.42 ± 0.08 a	14.24 ± 0.16 a
−	+	−	6.93 ± 0.09 c	2.81 ± 0.08 c	2.87 ± 0.11 c	12.61 ± 0.09 c
−	+	+	7.40 ± 0.14 b	3.01 ± 0.11 b	3.15 ± 0.18 b	13.60 ± 0.30 b
+	−	−	4.20 ± 0.11 e	1.87 ± 0.07 f	2.10 ± 0.04 e	8.17 ± 0.20 g
+	−	+	6.31 ± 0.22 d	2.94 ± 0.17 b	2.98 ± 0.13 c	12.23 ± 0.27 d
+	+	−	6.30 ± 0.04 d	2.57 ± 0.15 d	2.50 ± 0.16 d	11.37 ± 0.24 f
+	+	+	6.22 ± 0.16 d	2.77 ± 0.12 c	2.86 ± 0.07 c	11.85 ± 0.35 e

Pb = lead; DOP = dopamine; EBR = 24-epibrassinolide; LDM = leaf dry matter; RDM = root dry matter; SDM = stem dry matter and TDM = total dry matter. Columns with different letters indicate significant differences from the Scott-Knott test (*p* < 0.05). Means ± SD. n = 5.

## Data Availability

Data are available upon request to the corresponding author.

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
