# Peer review of "Oxidative Stress and Negative Consequences on Photosystem II Occasioned by Lead Stress Are Mitigated by 24-Epibrassinolide and Dopamine in Tomato Plants"

_plants, 2025, doi:10.3390/plants14233699_

Round 1

Reviewer 1 Report

Comments and Suggestions for Authors

In this study, the authors confirmed the positive roles of EBR (24-epibrassinolide) and DOP in mitigated the damage caused by Pb (lead) exposure in tomato plants. The results are interesting and showed the potential application in defense Pb stress in plants. However, several problems should be clarified before it is considered to be published.

  1. How are the exogenous application concentration of EBR and DOP?
  2. The author indicated that “Pb Concentrations Were Minimized After Treatment with DOP and EBR”. The phenotypes of plants under this condition should be showed.
  3. The author indicated that “Exogenous Spraying of DOP and EBR Maximized Gas Exchange”. The stomatal aperture in leaves should be measured.
  4. The author indicated “DOP and EBR Promoted Higher Biomass Accumulation in Plants Exposed to Pb Excess”. The phenotypes of plants should be showed in the manuscript.

Author Response

Dear reviewers and editor linked to manuscript # plants-3947437:

We are submitting the revised manuscript to the journal submission site for your review.

As instructed in your e-mail on 22 Out 2025, we have carefully considered all the reviewer’s comments and fully addressed them in the revised manuscript. Our responses to each specific reviewer comments are as follows (red into manuscript):

Decision from Editor: Your manuscript has now been reviewed by experts in the field and can be found with the review reports.

Decision: Major revisions

Ms. Lumi Li

Editor

Plants

General comments from Reviewer 1: In this study, the authors confirmed the positive roles of EBR (24-epibrassinolide) and DOP in mitigated the damage caused by Pb (lead) exposure in tomato plants. The results are interesting and showed the potential application in defense Pb stress in plants. However, several problems should be clarified before it is considered to be published.

Reviewer 1: How are the exogenous application concentration of EBR and DOP?

Authors: This information was inserted (below):

2.4. Dopamine (DOP) and 24-Epibrassinolide (EBR) Preparations and Applications

Dopamine (DOP) solution with 100 µM (Sigma-Aldrich, USA) was prepared [38]. A 100 nM 24-epibrassinolide (EBR) (Sigma-Aldrich, USA) solution was used [39]. Fifteen-day-old plants were sprayed with single and combined applications of DOP and EBR (pre-treatment) at 7-d intervals until day 50.

Reviewer 1: The author indicated that “Pb Concentrations Were Minimized After Treatment with DOP and EBR”. The phenotypes of plants under this condition should be showed.

Authors: We don't have images (plant). Our photographic equipment was malfunctioning during this research.

Reviewer 1: The author indicated that “Exogenous Spraying of DOP and EBR Maximized Gas Exchange”. The stomatal aperture in leaves should be measured.

Authors: This subtitle is adequate, as proven in this research. We measured stomatal conductance, transpiration rate, net photosynthetic rate, intercellular CO2 concentration, water-use efficiency, and carboxylation instantaneous efficiency.

Reviewer 1: The author indicated “DOP and EBR Promoted Higher Biomass Accumulation in Plants Exposed to Pb Excess”. The phenotypes of plants should be showed in the manuscript.

Authors: We don't have images (plant). Our photographic equipment was malfunctioning during this research.

General comments from Reviewer 2: This manuscript presents a well-designed and comprehensive study investigating the individual and combined effects of 24-epibrassinolide (EBR) and dopamine (DOP) in mitigating lead (Pb)-induced stress in tomato plants. Multiple physiological and biochemical parameters were evaluated, including nutrient uptake, photosynthetic performance, chlorophyll fluorescence, antioxidant enzyme activities, oxidative stress markers, and biomass accumulation. The study provides the first systematic comparison of EBR and DOP, both alone and in combination, in alleviating Pb toxicity in tomatoes. The findings offer potential strategies for enhancing crop resilience in heavy-metal-contaminated soils. Overall, the research is novel, experimentally sound, and addresses a relevant issue in plant stress physiology and agricultural sustainability. The manuscript is generally well-written, the data are robust, and the conclusions are supported by the results. I recommend minor revisions before acceptance.

Recommendation: Minor revision required. This manuscript makes a valuable contribution to the field of plant stress physiology. With the suggested revisions—particularly in methodological details, figure clarity, and mechanistic discussion—it will be suitable for publication.

Reviewer 2: Materials and Methods / Growth Conditions: Please specify the photoperiod and light intensity in the greenhouse to enhance reproducibility.

Authors: This information was included “with photosynthetic photon flux density (PPFD) oscillating between 1150 and 550 µmol m-2 s-1 (measured at 12 am), and 12-h photoperiod.”

Reviewer 2: Materials and Methods / Treatment Timing: Clarify the exact timing and frequency of EBR and DOP applications relative to Pb exposure (e.g., pre-, during, or post-stress).

Authors: More details were included (below):

2.4. Dopamine (DOP) and 24-Epibrassinolide (EBR) Preparations and Applications

Dopamine (DOP) solution with 100 µM (Sigma-Aldrich, USA) was prepared [38]. A 100 nM 24-epibrassinolide (EBR) (Sigma-Aldrich, USA) solution was used [39]. Fifteen-day-old plants were sprayed with single and combined applications of DOP and EBR (pre-treatment) at 7-d intervals until day 50.

2.5. Plant Nutrition and Pb Treatment

The plants received the following macro and micronutrients contained in the nu-trient

solution: 8.75 mM KNO3, 7.5 mM Ca(NO3)2·4H2O, 3.25 mM NH4H2PO4, 1.5 mM MgSO4·7H2O, 62.50 µM KCl, 31.25 µM H3BO3, 2.50 µM MnSO4·H2O, 2.50 µM ZnSO4·7H2O, 0.63 µM CuSO4·5H2O, 0.63 µMNaMoO4·5H2O, and 250.0 µM NaEDTA-Fe·3H2O. To induce Pb stress, PbCl2 was used at concentrations of 0 and 200 µM Pb and was applied over 10 days (days 40–50 after the start of the experiment). During the study, the nutrient solutions were changed at 07:00 h at 3-day intervals, with the pH adjusted to 5.5 using HCl or NaOH. On day 50 of the experiment, physiological and morphological parameters were measured for all plants, and tissues were harvested for anatomical, biochemical and nutritional analyses.

Reviewer 2: Materials and Methods / Statistical Details: Mention the version of the statistical software used and justify the choice of the Scott-Knott test for multiple comparisons.

Authors: This information were included (below):

2.21. Data Analysis

The data were subjected to an analysis of variance, and significant differences between the means were determined using the Scott-Knott test at a probability level of 5%, because this test better distinguishes the differences between averages. [56]. Standard deviations were calculated for each treatment. Statistical analysis of the data was done using R™ software version 4.4.1 [57].

Reviewer 2: Results / Data Consistency: Some percentage changes described in the text (e.g., "35%") do not fully align with the values in the tables. Please verify these calculations.

Authors: All data and percentages were checked and, when necessary, corrected. Now they are adequate.

Reviewer 2: Discussion / Mechanistic Insight: While the discussion covers possible mechanisms, it would benefit from a deeper exploration of the potential synergistic or antagonistic interactions between EBR and DOP. Consider discussing cross-talk in signaling pathways (e.g., hormone interactions, redox homeostasis).

Authors: Thank you very much for the suggestion. The discussion has been improved (synergistic interactions between EBR and DOP).

Reviewer 2: Discussion / Recent Literature: Incorporate more recent studies on the role of dopamine in plant stress responses to strengthen the context and relevance of the findings.

Authors: Recent and relevant studies were included.

Reviewer 2: Language and Formatting / Language Polish: Some sentences are overly long or complex. Consider revising for conciseness and readability.

Authors: Manuscript was again revised by a language service from a traditional publisher.

Reviewer 2: References: Ensure uniformity in reference formatting. Some entries are missing page numbers or DOIs.

Authors: References were checked and adjusted.

General comments from Reviewer 3: The study summarizes results of a controlled experiment on tomato plants, that were treated with lead (Pb) in parallel with applying different combinations of dopamine and epibrassinolide on the aerial parts. The Authors analyzed growth, ionomics, photosynthetic processes, oxidative markers and antioxidant defense of the plants and concluded that both bioactive compounds helped in alleviating Pb-stress.

In general the study fits into the scope of the journal Plants, its experimental design and methods are suitable to address the aims, and the results are presented and discussed in enough details. The text is well-written and easy to follow. Therefore, I have only minor remarks and suggestions to improve the study.

Reviewer 3: l 23: correctly "in Pb-stressed tomato"

Authors: Correction implemented.

Reviewer 3: l 28 and throughout the text: instead of "molecules", I would rather suggest using "compound", or something synonymous

Authors: Correction implemented.

Reviewer 3: l 34 (keywords): correctly it's "neurotransmitter", but I would rather name "dopamine" instead because you didn't use dopamine as a neurotransmitter, but as a bioactive compound in plants. The same stands for the other occurences, e.g. in l 321 and l 425

Authors: Correction implemented.

Reviewer 3: l 56: neither element is biodegradable, you may omit either "non-biodegradable" or "element"

Authors: Correction implemented.

Reviewer 3: l 116: please list the nurtients with the respective concentrations. This sentence looks odd: "The plants received the following macro and micronutrients [40]."

Authors: Correction implemented.

2.5. Plant Nutrition and Pb Treatment

The plants received the following macro and micronutrients contained in the nu-trient

solution: 8.75 mM KNO3, 7.5 mM Ca(NO3)2·4H2O, 3.25 mM NH4H2PO4, 1.5 mM MgSO4·7H2O, 62.50 µM KCl, 31.25 µM H3BO3, 2.50 µM MnSO4·H2O, 2.50 µM ZnSO4·7H2O, 0.63 µM CuSO4·5H2O, 0.63 µMNaMoO4·5H2O, and 250.0 µM NaEDTA-Fe·3H2O. To induce Pb stress, PbCl2 was used at concentrations of 0 and 200 µM Pb and was applied over 10 days (days 40–50 after the start of the experiment). During the study, the nutrient solutions were changed at 07:00 h at 3-day intervals, with the pH adjusted to 5.5 using HCl or NaOH. On day 50 of the experiment, physiological and morphological parameters were measured for all plants, and tissues were harvested for anatomical, biochemical and nutritional analyses.

Reviewer 3: l 137: Did I get it right that you measured light-acclimated chlorophyll fluorescence parameters under ambient light? If so, please provide a PPFD range that the leaves were exposed to during the measurements. If not, please state the actinic light intensity. How were the leaves dark-adapted? In which period of the day were the measurements performed?

Authors: This information was added (below):

The chlorophyll fluorescence was measured in fully expanded leaves under light between 10:00 and 12:00 h. Preliminary tests determined that the acropetal third of leaves in the middle third of the plant and that adapted to the dark for 30 min yielded the greatest Fv/Fm ratio. Therefore, this part of the plant was used for measurements. The intensity and duration of the saturation light pulse were 7,500 µmol m–2.s–1 and 0.7 s, respectively.

Reviewer 3: l 175: typo in "μmol AsAmg–1 protein min–1."

Authors: Correction implemented.

Reviewer 3: l 207: "Electrolyte leakage was measured with minor modifications" - minor modifications compared to what?

Authors: Correction implemented (below):

2.18. Determination of Electrolyte Leakage

Electrolyte leakage was measured with minor modifications [54]. Fresh tissue (200 mg) was cut into pieces 1 cm in length and placed in containers with 8 mL of distilled deionized water. The containers were incubated in a water bath at 40 °C for 30 min and the initial electrical conductivity of the medium (EC1) was measured. The samples were then boiled at 95 °C for 20 min to release the electrolytes. After cooling, the final electrical conductivity (EC2). The percentage of electrolyte leakage was calculated using the formula EL (%) = (EC1/EC2) × 100.

Reviewer 3: l 228: If you used R, please state its version too

Authors: This information were included (below):

2.21. Data Analysis

The data were subjected to an analysis of variance, and significant differences between the means were determined using the Scott-Knott test at a probability level of 5%, because this test better distinguishes the differences between averages. [56]. Standard deviations were calculated for each treatment. Statistical analysis of the data was done using R™ software version 4.4.1 [57].

Reviewer 3: Table 3: The Chl/Car ratio seems suspiciously high. Normally, it ranges around 3. Please double-check the calculations for Car content, or the accuracy of your method, wavelengths, etc.

Authors: All data and percentages were checked and, when necessary, corrected. Now they are adequate.

Reviewer 3: Throughout the Results section: I would suggest omitting discussion-like statements in the subsection titles, such as e.g.: "3.1. Pb Concentrations Were Minimized After Treatment with DOP and EBR"

Authors: We appreciate the reviewer 3 suggestion, opinion, and time spent reviewing this research, but we believe the subheadings are appropriate.

Reviewer 3: l 464: I didn't check the source study, but "643%" seems to be a typo

Authors: Sentence was corrected.

Reviewer 3: l 508-512: It's a general description about Pb toxicity. I wouldn't suggest placing it to the part where you discuss the effects of EBR and DOP, and especially not as a closing section of your Discussion.

Authors: We appreciate the reviewer 3 suggestion, opinion, and time spent reviewing this research, but we believe that this sentence is appropriate (Pb impacts on pigments, photosynthesis, and biomass).

Reviewer 3: Just a suggestion: I know that it'd be quite a lot extra work, but you may consider using multivariate analyses (correlation matrices, PCA), and/or two-way ANOVA, and present your results based on those.

Authors: We appreciate the suggestion; however, we are unsure how to perform this analysis.

We thank you again for your time and effort in handling and reviewing our manuscript and we are looking forward to hearing from you.

Sincerely yours,

Allan Klynger da Silva Lobato

Professor / Universidade Federal Rural da Amazônia

Affiliate member / Brazilian Academy of Sciences

+55 91 993134006

allan.lobato@ufra.edu.br

Reviewer 2 Report

Comments and Suggestions for Authors

This manuscript presents a well-designed and comprehensive study investigating the individual and combined effects of 24-epibrassinolide (EBR) and dopamine (DOP) in mitigating lead (Pb)-induced stress in tomato plants. Multiple physiological and biochemical parameters were evaluated, including nutrient uptake, photosynthetic performance, chlorophyll fluorescence, antioxidant enzyme activities, oxidative stress markers, and biomass accumulation. The study provides the first systematic comparison of EBR and DOP, both alone and in combination, in alleviating Pb toxicity in tomatoes. The findings offer potential strategies for enhancing crop resilience in heavy-metal-contaminated soils. Overall, the research is novel, experimentally sound, and addresses a relevant issue in plant stress physiology and agricultural sustainability. The manuscript is generally well-written, the data are robust, and the conclusions are supported by the results. I recommend minor revisions before acceptance.

  1. Materials and Methods

- Growth Conditions: Please specify the photoperiod and light intensity in the greenhouse to enhance reproducibility.

- Treatment Timing: Clarify the exact timing and frequency of EBR and DOP applications relative to Pb exposure (e.g., pre-, during, or post-stress).

- Statistical Details: Mention the version of the statistical software used and justify the choice of the Scott-Knott test for multiple comparisons.

  1. Results

- Data Consistency: Some percentage changes described in the text (e.g., "35%") do not fully align with the values in the tables. Please verify these calculations.

  1. Discussion

- Mechanistic Insight: While the discussion covers possible mechanisms, it would benefit from a deeper exploration of the potential synergistic or antagonistic interactions between EBR and DOP. Consider discussing cross-talk in signaling pathways (e.g., hormone interactions, redox homeostasis).

- Recent Literature: Incorporate more recent studies on the role of dopamine in plant stress responses to strengthen the context and relevance of the findings.

  1. Language and Formatting

- Language Polish: Some sentences are overly long or complex. Consider revising for conciseness and readability.

- References: Ensure uniformity in reference formatting. Some entries are missing page numbers or DOIs.

Recommendation

Minor Revision Required. This manuscript makes a valuable contribution to the field of plant stress physiology. With the suggested revisions—particularly in methodological details, figure clarity, and mechanistic discussion—it will be suitable for publication.

Author Response

(The authors gave the same response as above.)

Reviewer 3 Report

Comments and Suggestions for Authors

The study summarizes results of a controlled experiment on tomato plants, that were treated with lead (Pb) in parallel with applying different combinations of dopamine and epibrassinolide on the aerial parts. The Authors analyzed growth, ionomics, photosynthetic processes, oxidative markers and antioxidant defense of the plants and concluded that both bioactive compounds helped in alleviating Pb-stress.

In general the study fits into the scope of the journal Plants, its experimental design and methods are suitable to address the aims, and the results are presented and discussed in enough details. The text is well-written and easy to follow.
Therefore, I have only minor remarks and suggestions to improve the study.

Detailed comments:
l 23: correctly "in Pb-stressed tomato"
l 28 and throughout the text: instead of "molecules", I would rather suggest using "compound", or something synonymous
l 34 (keywords): correctly it's "neurotransmitter", but I would rather name "dopamine" instead because you didn't use dopamine as a neurotransmitter, but as a bioactive compound in plants. The same stands for the other occurences, e.g. in l 321 and l 425
l 56: neither element is biodegradable, you may omit either "non-biodegradable" or "element"
l 116: please list the nurtients with the respective concentrations. This sentence looks odd: "The plants received the following macro and micronutrients [40]."
l 137: Did I get it right that you measured light-acclimated chlorophyll fluorescence parameters under ambient light? If so, please provide a PPFD range that the leaves were exposed to during the measurements. If not, please state the actinic light intensity. How were the leaves dark-adapted? In which period of the day were the measurements performed?
l 175: typo in "μmol AsAmg–1 protein min–1."
l 207: "Electrolyte leakage was measured with minor modifications" - minor modifications compared to what?
l 228: If you used R, please state its version too
Table 3: The Chl/Car ratio seems suspiciously high. Normally, it ranges around 3. Please double-check the calculations for Car content, or the accuracy of your method, wavelengths, etc.
Throughout the Results section: I would suggest omitting discussion-like statements in the subsection titles, such as e.g.: "3.1. Pb Concentrations Were Minimized After Treatment with DOP and EBR"
l 464: I didn't check the source study, but "643%" seems to be a typo
l 508-512: It's a general description about Pb toxicity. I wouldn't suggest placing it to the part where you discuss the effects of EBR and DOP, and especially not as a closing section of your Discussion.

Just a suggestion: I know that it'd be quite a lot extra work, but you may consider using multivariate analyses (correlation matrices, PCA), and/or two-way ANOVA, and present your results based on those.

Author Response

(The authors gave the same response as above.)

Round 2

Reviewer 1 Report

Comments and Suggestions for Authors

The authors have solve most of the questions except the author explain that"We don't have images (plant). Our photographic equipment was malfunctioning during this research"and the langugage of the presnt MS was satisfactory